# Long-Term Outcomes and Predictive Factors of Hospitalized Patients with Severe Ulcerative Colitis Treated with Intravenous Corticosteroids

**DOI:** 10.3390/jcm10225413

**Published:** 2021-11-19

**Authors:** Elena De Cristofaro, Silvia Salvatori, Irene Marafini, Francesca Zorzi, Norma Alfieri, Martina Musumeci, Livia Biancone, Emma Calabrese, Giovanni Monteleone

**Affiliations:** Department of Systems Medicine, University of Rome “Tor Vergata”, 00133 Rome, Italy; elena_decr@hotmail.it (E.D.C.); silviasalvatori23@gmail.com (S.S.); irene.marafini@gmail.com (I.M.); fra.zorzi@yahoo.it (F.Z.); norma.alfieri@outlook.com (N.A.); martinamusu92@gmail.com (M.M.); biancone@med.uniroma2.it (L.B.); emma.calabrese@uniroma2.it (E.C.)

**Keywords:** inflammatory bowel disease, acute severe UC, steroids

## Abstract

*Background and Aims*: Treatment with intravenous corticosteroids (IVCS) is a mainstay in the management of acute severe ulcerative colitis (UC). Although most patients respond to IVCS, little is known about the long-term outcomes. In this study, we assessed the long-term outcomes of IVCS in a real-life cohort. *Methods*: Disease activity, clinical relapse (partial Mayo score >4), the need for steroids or other maintenance therapies and the rates of colectomy and re-hospitalization were evaluated in consecutive patients admitted to the Tor Vergata University hospital between 2010 and 2020 for acute severe UC who responded to IVCS. *Results:* Eighty-eight patients were followed up with for a median period of 46 (range 6–133) months. Of these, 56 (64%) patients were treated with 5-aminosalycilic acid and 32 (36%) with immunomodulators or biologics after discharge. A total of 60 out of 88 patients (68%) relapsed, 28 (32%) were re-hospitalized, and 15 (17%) underwent a colectomy with no difference between the two maintenance therapy groups. The multivariate analysis showed that patients in clinical remission 6 months after discharge had a lower risk of relapse during the follow-up. *Conclusions:* Nearly two-thirds of patients with acute UC responding to IVCS experienced relapse after a median follow-up of 4 years, and this was not influenced by the maintenance therapy.

## 1. Introduction

Ulcerative colitis (UC) is a chronic immune-mediated disorder of the gastrointestinal tract of unknown etiology in which inflammation starting from the rectum can extend proximally and involve the whole colon [1]. Nowadays, UC prevalence varies from 37.5 to 238/100,000 and from 21 to 294/100,000 in North America and Europe, respectively [2,3]. Acute severe UC is a life-threatening condition, which requires hospitalization of the patients [4,5]. The cornerstone of acute severe UC treatment remains intravenous corticosteroids (IVCS), the use of which, together with fluid and electrolyte resuscitation, nutritional support, and thromboprophylaxis, has significantly decreased mortality among this subgroup of patients [6,7]. However, nearly one-third of patients with acute severe UC do not respond to IVCS [6,8]. Several authors have analyzed the factors associated with the lack of response to IVCS and colectomy [8,9]. For instance, it was demonstrated that patients with more than eight bowel movements per day or with serum levels of C-reactive protein higher than 45 mg/L on the third day of IVCS treatment have high risk of colectomy [10,11,12]. Similarly, the presence of severe endoscopic lesions was associated with an increased risk of IVCS treatment failure [13]. A systematic review of 32 cohort studies and controlled trials published between 1974 and 2006 identified more than 20 predictors of IVCS failure in the adult UC population, but only a few parameters, such as disease extent, stool frequency, temperature, heart rate, C-reactive protein (CRP), albumin, and radiologic assessment (mucosal tags and bowel dilatation) were consistently reproduced [6]. Second-line therapy with cyclosporine or infliximab is a rescue treatment for those patients who do not respond to IVCS after 3–5 days [14], and this has improved the prognosis of patients with IVCS-refractory UC, mainly in terms of the colectomy rate. In contrast, little is known about the long-term outcomes of patients responding to IVCS during hospitalization for severe relapse. A recent retrospective, multi-center study evaluated the relapse-free survival in 142 patients with acute severe UC responding to IVCS and followed up with four academic inflammatory bowel disease (IBD) centers. A high relapse rate together with a low rate of colectomy after 5 years of follow-up was documented. Early response to IVCS and maintenance therapy with biologics were associated with a lower rate of relapse [15]. To explore this issue further, we assessed the long-term outcomes of a real-life cohort of hospitalized patients with acute severe UC responding to IVCS and investigated the predictive factors of relapse.

## 2. Materials and Methods

### 2.1. Study Population and Data Collection

This retrospective study included all patients with established UC diagnosis admitted to the Tor Vergata University hospital between 2010 and 2020 for severe relapse who were successfully treated with IVCS. In all cases, hospitalization was due to the first severe exacerbation of colitis and, therefore, all the patients received the first course of IVCS treatment. UC clinical activity was defined according to the modified Truelove and Witts criteria [16]. Exclusion criteria included: mild or moderate UC, evidence of intestinal superinfection with Cytomegalovirus, Clostridium difficile or other intestinal pathogens, a diagnosis of Crohn’s disease or unclassified IBD, and failure of IVCS leading to colectomy and/or second-line therapy. Patients were recruited from the standardised hospital in-patient diagnostic dataset by searching for the International Classification of Diseases (ICD-10) codes (556.0, 556.3, 556.6, 556.8, 556.9). Demographic, clinical and endoscopic characteristics of the patients were collected from medical records and included sex, age, smoking habits, history of medical and surgical treatment of UC, date of diagnosis, disease behaviour prior to admission, and the dates of admission and discharge. Flexible sigmoidoscopy was performed at baseline in all patients. Data about disease extent, which was evaluated according to the Montreal classification, were collected from medical records. Disease activity was assessed at baseline using partial Mayo Clinic scores for clinical activity [17,18], CRP (mg/L), hemoglobin (g/dL), and albumin (g/dL), and for laboratory activity and Mayo Clinic endoscopic sub-scores for endoscopic activity [17,18]. Intensive medical therapy consisted of intravenous methylprednisolone at a dose 60 mg per day with fluid and electrolyte resuscitation and thromboprophylaxis with subcutaneous low-molecular-weight heparin. Response to IVCS was defined as a resolution of clinical symptoms with ≤3 stools per day without visible bleeding. Refractoriness to IVCS was defined according to clinical and biochemical criteria at days 3 and 5. Concomitant use of rectal 5-ASA and/or CS formulations was permitted during hospitalization. The maintenance therapy at discharge was not standardized and was at the physician’s discretion. The protocol was approved by the local Ethics Committee (N. 217.21).

### 2.2. Outcome Measurement during Follow-Up

After discharge, the patients were evaluated for disease activity, treatment, and adverse events at 6, 12, and 24 months, and at the end of the follow-up corresponding to their last visit or colectomy. The follow-up time points slightly varied among patients, and not all of them were seen at all the specified time points. Disease activity was assessed during follow-up using partial Mayo Clinic scores [17,18]. The need for steroids, biological therapies, hospitalization, and colectomy was evaluated in the follow-up period. Clinical remission was defined as a partial Mayo score of ≤2 with a combined stool frequency and rectal bleeding sub-score of ≤1. Steroid-free clinical remission was defined as clinical remission without any oral steroid treatment. Relapse was defined as active disease with a partial Mayo Clinic score of >4. At least one year after their discharge, a colonoscopy was performed in patients complaining of clinical symptoms. Endoscopic remission was defined as Mayo Clinic endoscopic sub-score of 0–1.

### 2.3. Statistical Analysis

All patients were evaluated from discharge to the end of the follow-up period. Qualitative data were expressed as numbers and proportions (%), and quantitative data were expressed as mean ± SD or median and interquartile range (IQR). Patients’ characteristics were compared using the χ^2^ or Fisher’s exact test for the categorical variables and the Mann–Whitney U test for continuous data. The paired *t*-test was used for evaluating the endoscopic changes 1 year after discharge. The relapse-free survivals were calculated using the Kaplan–Meier method. To determine the risk factors of relapse, a Cox proportional hazard model was performed. The variables with a significant *p* value of <0.05 in univariate analysis were then considered potential control variables for multivariate analysis. 

All analyses were two-tailed, and *p*-value < 0.05 was considered statistically significant. Statistical analyses were performed using a GraphPad Prism (software version 9.0.0, San Diego, CA, USA).

## 3. Results

### 3.1. Study Population

A total of 141 hospital in-patients with severe relapse were screened for inclusion. Of these, 17 patients required rescue therapy with Infliximab, and 11 patients underwent urgent colectomy due to the failure of IVCS therapy and previous lack of response to anti-TNFs. Among the 113 patients responding to IVCS therapy, 25 were lost in the early phases of the follow-up. Therefore, 88 patients were included in the study (Figure 1).

### 3.2. Characteristics at Baseline of Patients Responding to IVCS and Maintenance Therapies

A total of 47 out of the 88 patients (53%) was male, 28 (32%) had a left-sided colitis, and 60 (68%) had an extensive colitis. The median duration of disease was 126 months (range 6–444). A total of 28 out of the 88 (32%) patients had a history of steroid dependence, and 14 patients (16%) had been exposed to anti-TNFα before admission to our hospital. Of the 88 patients, 83 (94%) had positive CRP values (>5 mg/L), 17 (19%) had hemoglobin values <10 g/dL, and 27 (31%) had hypoalbuminemia (albumin <3 g/dL) upon hospital admission. The Mayo Clinic endoscopic sub-score showed severe activity in 69 out of 88 patients (78%); the remaining 19 (22%) had moderate activity in the distal colon, but they were receiving rectal 5-ASA upon hospital admission. The flow chart of patients according to maintenance therapy prescribed during the first three months after discharge is shown in Figure 1. A total of 56 out of 88 patients (64%) were treated with 5-ASA, and 32 (36%) received immunomodulators (IMMs: azathioprine or 6-mercaptopurine (*n* = 12)) or biologics (*n* = 20) (anti-TNFs, *n* = 17; vedolizumab, *n* = 3). IMM/biologics were prescribed as maintenance therapy in patients who were steroid dependent (22 patients), had concomitant extra-intestinal manifestation (3 patients), or had had more than two relapses during the last year (7 patients). The clinical and demographic characteristics of the 88 patients are shown in Table 1. Patients with previous exposure to biological agents were less frequently treated with 5-ASA (hazard ratio (HR) = 0.31; 95% CI: 0.092–0.99; *p* = 0.04) and more frequently with IMM/biological therapies (HR = 3.261, 95% CI: 1.004–10.83; *p* = 0.004). Similarly, steroid-dependent patients received IMM/biological agents more frequently (HR = 6.31, 95% CI: 2.32–15.4; *p* = 0.0002) and 5-ASA less frequently (HR = 0.16, 95% CI: 0.065–0.43; *p* = 0.0002). No other differences in demographic, clinical, or biochemical findings were seen between the two maintenance therapy groups, except for serum albumin levels at admission, which were lower in the 5-ASA group as compared with the IMM/biologic group (*p* = 0.0048) (Table 1).

### 3.3. Long-Term Outcomes in Patients Responding to IVCS

Table 2 shows the long-term outcomes of the 88 patients achieving clinical remission following IVCS therapy. During a median follow-up of 46 (range: 6–133) months, 60 of 88 patients (68%), 39 out of 56 (70%) in the 5-ASA group, and 21 out of 32 (66%) in IMM/biologic group (*p* = 0.81) relapsed. Among the relapsers, 49 (82%) patients, 32 out of 39 (82%) in the 5-ASA group, and 17 out of 21 (81%) in the IMM/biologic group (*p* = 0.82) needed steroids. Half of the patients (51%) had more than one exacerbation during the follow-up period. Steroid-free clinical remission was documented in 24 patients (43%) in the 5-ASA group and 14 (44%) patients in the IMM/biologic group (*p* = 0.83). A total of 28 out of 88 patients (32%) were re-hospitalized for a severe clinical relapse of the disease (17 in the 5-ASA group and 11 in the IMM/biologic group, *p* = 0.81) and 15 out of 88 patients (17%) underwent a colectomy (11 in the 5-ASA group and 4 in IMM/biologic group; *p* = 0.56). The use of other biological therapies was significantly higher in the IMM/biologic group (16/31 patients) compared with the 5-ASA group (15/56 patients) (*p* = 0.037). Further analysis of these outcomes at earlier time points showed that nearly one-third of the patients relapsed as early as 6 months after discharge, and, overall, one fifth of the patients received at least one more cycle of systemic steroids at the same time point (Table 3). The univariate analysis demonstrated that the risk of relapse was significantly increased in patients with steroid-dependent history and in those who had been previously exposed to anti-TNF, while it was significantly decreased in patients who were in clinical remission six months after discharge (Table 4). However, the multivariate analysis showed that only clinical remission at month 6 after discharge was a significant protective factor of relapse (HR 0.22; 95% CI: 0.05–0.88, *p* = 0.03) (Table 4). The probabilities of relapse-free survival were 67%, 33%, and 24% in the 5-ASA group and 54%, 27%, and 21% in the IMM/biologic group, respectively, at 2, 4, and 6 years (*p* = 0.96) (Figure 2). Data about the endoscopic activity one year after discharge were available for 47 of the 88 patients (53%). At this time point, there was a significant improvement in Mayo endoscopic sub-scores as compared to sub-scores documented at the time of hospitalization (Figure 3). This was evident in both the 5-ASA group (28 patients) (median values: 2.8 and 2.0 at the time of hospitalization and 1 year after discharge, respectively, *p* = 0.0035) and IMM/biologic group (19 patients) (median values: 2.7 and 1.4 at the time of hospitalization and 1 year after discharge, respectively, *p* = 0.0002).

## 4. Discussion

The present study focused on the long-term outcomes and predictive factors of relapse in a cohort of hospitalized in-patients with acute severe UC who responded to IVCS treatment. After a median follow-up of 4 years, nearly two-thirds of the patients relapsed, and the relapse-free survival rate was 25% after 6 years. Such results were independent of the maintenance therapy adopted and confirmed even when a comparison was made between patients receiving only TNF blockers and those treated with 5-ASA. Our findings, in part, conflict with those published by Salameh and colleagues, who documented that most patients experienced clinical relapses due to the failure of the chosen maintenance strategy, and a lower rate of relapses was documented in patients treated with anti-TNF as compared to those receiving 5-ASA or immunomodulators [15]. Factors accounting for such a discrepancy remain unknown, even though they could, at least in part, rely on the small number of patients treated with TNF blockers after discharge in both studies (18/142 and 17/88, respectively). One-third of our patients relapsed as early as 6 months after discharge. Since responsiveness to IVCS was ascertained only clinically, because no endoscopic/histologic data were available, it is conceivable that the early clinical flare-up was related to an incomplete response to IVCS. Another possibility is that early relapse occurs in a subgroup of patients with a more aggressive course. Indeed, our data indicate that more than two-thirds of the relapses needed treatment with steroids, and re-hospitalization and colectomies at the same time point occurred only in this subgroup of patients. Recently, ECCO guidelines recommended an accelerated step-up strategy in patients with acute severe UC, using thiopurines for naïve patients and biological agents in thiopurine refractory patients [19], even though the primary use of biologics and/or immunomodulator therapy vs. gradual step-up therapy has not yet been evaluated in patients with moderate–severe UC. In this context, it is also noteworthy that, in a retrospective study including 185 patients, Vedamurthy et al. [20] examined the benefits of early therapeutic escalation in immunosuppression-naïve patients hospitalized with acute severe UC responsive to corticosteroids. The authors documented no benefit of early immunosuppression on the risk of colectomy, but the study did not include patients treated with biological agents. At the same time, other studies assessing the long-term follow-up of patients treated with 5-ASA after initial disease flare-up showed that 5-ASA is an effective maintenance therapy after the first steroid course [21,22]. Up to one-third of UC patients may require colectomy for the treatment of their disease. It has also been shown that patients who require hospitalization for the management of disease flare-ups are at a higher risk of needing a subsequent colectomy during the course of their disease [23]. Long-term follow-ups after cyclosporine-induced remission of severe UC showed that one-third of patients required colectomy at 1 year, while more than two-thirds underwent surgery at 5 years [24]. In our study, the rate of colectomy was 17%, which is in agreement with the results published by Salameh and colleagues, who showed that the probabilities of colectomy were 9% and 12% at 5 and 8 years, respectively. Our data also show that the type of maintenance therapy did not influence the rate of colectomy. This is consistent with the results published by Vedamurthy and colleagues who followed up with 133 immunosuppressive-naïve patients hospitalized for UC. Of these, 13 patients who responded to IVCS and did not require rescue therapy underwent colectomy by 1 year, with no difference being observed between those treated with immunomodulators and with 5-ASA [20]. A prospective study, performed by Bojic and colleagues in patients who avoided colectomy after receiving intensive treatment for severe UC, showed that patients with an incomplete response had a greater chance of colectomy in the long-term follow-up, despite optimal treatment with cyclosporin and azathioprine [25]. The same study showed that the maximum duration of remission was longer in patients with a complete response to IVCS than in patients with an incomplete response. These latter results are in line with our multivariate analysis, which showed that maintaining remission in the first six months after discharge is a protective factor of relapse. Consistently, the percentage of relapse was greater in patients with endoscopic remission at 1 year after discharge as compared with those with moderate/severe endoscopic activity.

We are aware that our study has some limitations. The retrospective nature of the study and the small number of patients receiving the maintenance therapies limit the applicability of the major findings and suggest the need for further and prospective studies. Second, the choice of the maintenance therapy was at the gastroenterologist’s discretion even though, in each case, patients were treated taking into account the previous clinical history, including steroid dependence and exposure to IMMs and biologics. During the follow-up, the definition of remission of the disease was mainly based on clinical symptoms/signs in the patients, because data about fecal calprotectin and endoscopic activity of the disease were lacking in nearly half of the patients. Our study also has some strengths. We recruited all UC patients hospitalized for severe relapse from the hospital dataset, which allowed us to generate reproducible results in a real-world population. Moreover, these data came from a single referral center with longstanding expertise in the management of UC. Finally, patients were monitored for a long period and factors influencing the outcomes were carefully analyzed.

## 5. Conclusions

In conclusion, although 80% of patients with acute severe UC respond to IVCS, after a median follow-up of 46 months, there is a high relapse rate regardless of maintenance therapy. Maintaining clinical remission in the first six months after discharge seems to be a protective factor for future relapses. Further longitudinal studies are required to identify predictive factors, which help clinicians to optimize treatment in patients who have a greater risk of relapse.

## Figures and Tables

**Figure 1 jcm-10-05413-f001:**
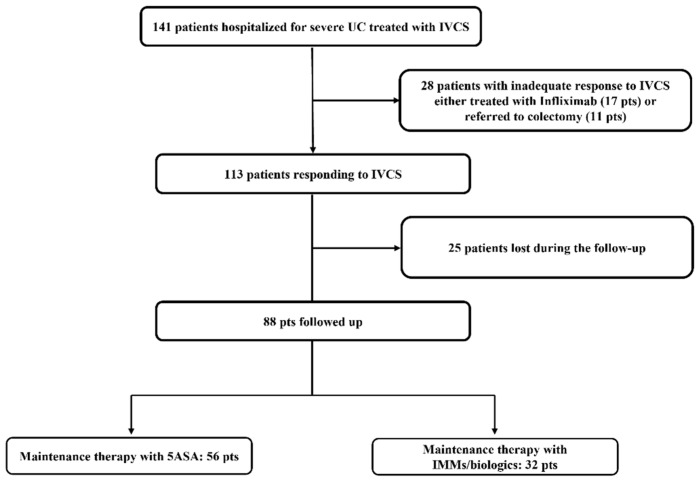
Flow-chart of patients with acute severe ulcerative colitis (UC) admitted to the hospital and initially treated with intravenous corticosteroids (IVCS). 5-ASA = 5-aminosalycilic acid; IMMs = immunosuppressors.

**Figure 2 jcm-10-05413-f002:**
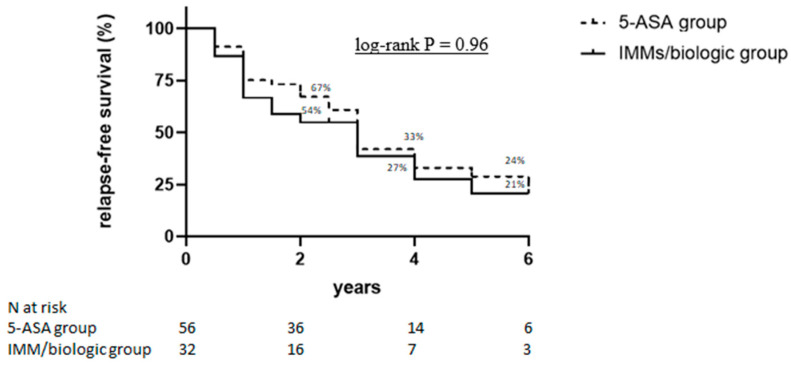
Kaplan–Meier curves of 88 patients with acute severe UC responding to intravenous corticosteroids (IVCS). Data about the relapse-free survival during the follow-up are shown according to maintenance therapy. 5-ASA = 5-aminosalycilic acid; IMMs = immunosuppressors.

**Figure 3 jcm-10-05413-f003:**
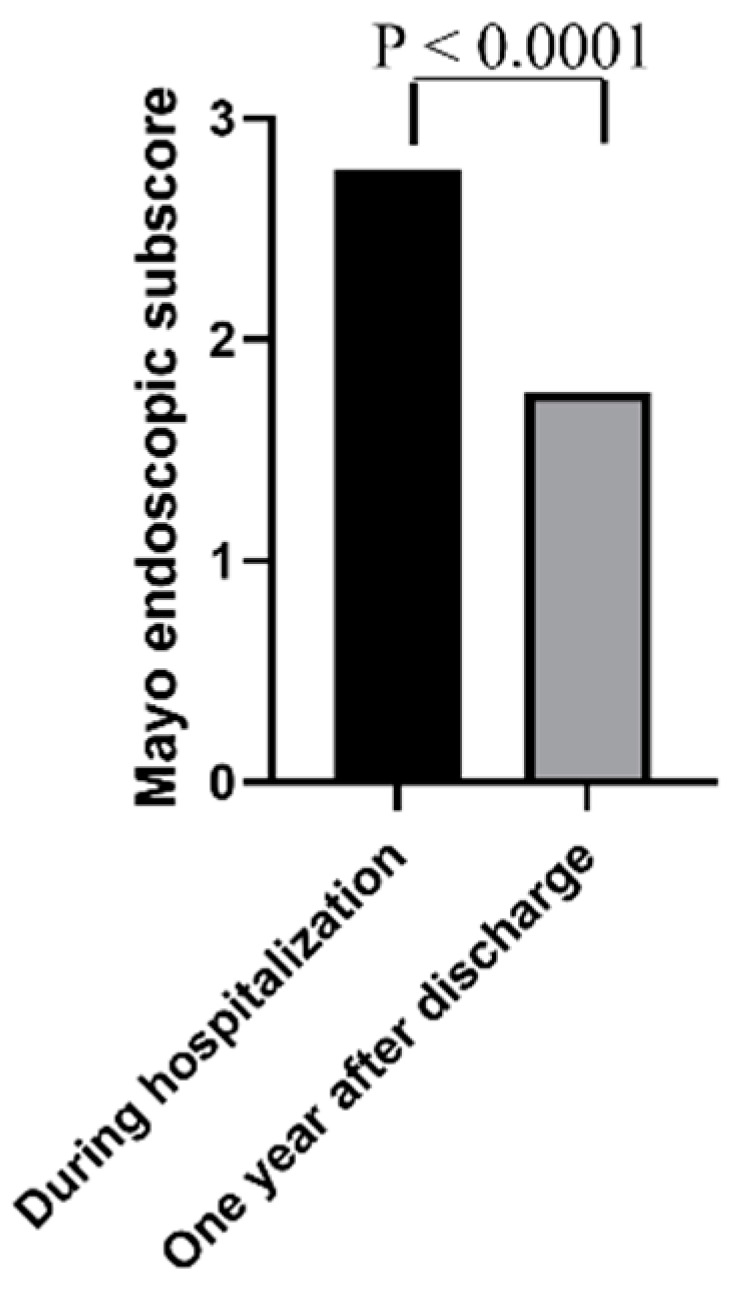
Mayo endoscopic sub-scores at the time of hospitalization and one year after discharge in 47 patients responding to intravenous corticosteroids (IVCS) and receiving 5-aminosalycilic acid (5-ASA) or immunosuppressors/biologics as maintenance therapy.

**Table 1 jcm-10-05413-t001:** Demographic and clinical characteristics at baseline of 88 patients with ulcerative colitis treated with intravenous corticosteroids. Patients were divided into 2 groups according to the maintenance therapy. 5-ASA = 5-aminosalycilic acid; IMM = immunosuppressor. Significant value is in bold.

Characteristics	Maintenance Therapy with 5-ASA (*n* = 56)	Maintenance Therapy with IMM/Biologic (*n* = 32)	*p*
Male gender, *n* (%)	33 (59%)	14 (44%)	0.19
Age, y (mean ± SD)	52 ± 19.0	44 ± 14.2	0.08
Appendectomy			
Yes, *n* (%)	3 (5%)	3 (9%)	0.056
Age at diagnosis:			
A2: 17–40 y	31 (55%)	16 (50%)	0.66
A3: >40 y	25 (45%)	16 (50%)	0.66
Ulcerative colitis			
E2: left-sided colitis	20 (36%)	8 (25%)	0.35
E3: extensive colitis	36 (64%)	24 (75%)	0.35
Duration of disease, months—median (IQR)	132 (108)	108 (138)	0.46
Previous anti-TNF therapies	6 (11%)	8 (25%)	**0.044**
Smoking habits			
Former	18 (32%)	14 (44%)	0.74
Current	7 (13%)	6 (19%)	0.34
Steroid dependence,			
Yes, *n* (%)	13 (23%)	15 (47%)	**0.032**
Partial Mayo Clinic score, (mean ± SD)	6.7 ± 1.1	6.6 ± 0.8	0.79
Endoscopic Mayo Clinic subscore,			
(mean ± SD)	2.6 ± 0.6	2.6 ± 0.5	0.79
C-reactive-protein (CRP)			
Positive (>5 mg/L), *n* (%)	55 (98%)	28 (87%)	0.056
Hemoglobin g/dL			
median (IQR)	11. 9 (2.8)	11. 7 (2.3)	0.909
Albumin g/dL			
median (IQR)	3 (0.7)	3.5 (0.8)	**0.0048**

**Table 2 jcm-10-05413-t002:** Long-term outcomes in 88 patients with severe ulcerative colitis treated with intravenous corticosteroids and different maintenance therapy. 5-ASA = 5-aminosalycilic acid; IMM = immunosuppressor. Significant value is in bold.

Long-Term Outcomes, *n* pts (%)	Maintenance Therapy with 5-ASA (*n* = 56)	Maintenance Therapy with IMM/Biologic (*n* = 32)	*p*
Relapse, *n* (%)			
(partial Mayo score >4),	39 (70%)	21 (66%)	0.81
Steroid-free clinical remission	24 (43%)	14 (44%)	0.83
Re-hospitalization, *n* (%)	17 (30%)	11 (34%)	0.81
Colectomy, *n* (%)	11 (20%)	4 (13%)	0.56
Corticosteroids, *n* (%)	32 (57%)	17 (53%)	0.82
IMMs, *n* (%)	9 (16%)	3 (9%)	0.52
Biologic therapy, *n* (%)	15 (27%)	16 (50%)	**0.037**

**Table 3 jcm-10-05413-t003:** Outcome measures of 88 patients with severe ulcerative colitis responding to intravenous corticosteroids during the first 2 years after discharge. IMMs = immunosuppressors.

	Month 6 (73 pts)	Month 12 (62 pts)	Month 24 (52 pts)
Relapse (partial Mayo score >4), *n* (%)	24 (33%)	18 (29%)	19 (36%)
Steroid-free clinical remission, *n* (%)	44 (60%)	41(66%)	33 (63%)
Re-hospitalization, *n* (%)	4 (5%)	6 (10%)	4 (8%)
Colectomy, *n* (%)	2 (3%)	0	0
Corticosteroids, *n* (%)	17 (23%)	14 (23%)	12 (23%)
IMMs, *n* (%)	1 (1.4%)	11 (18%)	7 (13%)
Biologic therapy, *n* (%)	6 (8%)	5 (8%)	5 (10%)

**Table 4 jcm-10-05413-t004:** Predictive variables associated with clinical relapse in 88 patients with acute severe ulcerative colitis responding to intravenous corticosteroids. IMMs = immunosuppressors. Significant value is in bold.

	Univariate Analysis		Multivariate Analysis	
Risk Factors	HR (95% CI)	*p* Value	HR (95% CI)	*p* Value
5-ASA maintenance therapy	1.01 (0.39 to 2.57)	0.99		-
IMM/biologic maintenance therapy	0.85 (0.34 to 2.16)	0.74		-
Previous anti-TNF exposure	8.71 (1.85 to 69.93)	**0.008**	6.51 (0.67 to 62.86)	0.89
Steroid dependence	3.04 (1.13 to 8.19)	**0.02**	1.09 (0.33 to 3.53)	0.11
Partial Mayo Clinic score <2 at 6 months after discharge	0.20 (0.05 to 0.76)	**0.02**	0.22 (0.05 to 0.88)	**0.03**
Disease duration (months)	0.99 (0.99 to 1.01)	0.78		-
Smoking habits	0.77 (0.23 to 2.59)	0.6		-

## Data Availability

The data that support the findings of this study are available from the corresponding author (G.M.), upon reasonable request.

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
