# Peer review of "Long-Term Outcomes and Predictive Factors of Hospitalized Patients with Severe Ulcerative Colitis Treated with Intravenous Corticosteroids"

_jcm, 2021, doi:10.3390/jcm10225413_

Round 1

Reviewer 1 Report

This article is interesting because it provides information on the long-term follow-up of patients with Ulcerative Colitis on Intravenous corticosteroid therapy.

  1. What is the prevalence of Ulcerative colitis? Would you please enter references indicating the prevalence of this disease?
  2. The material and methods stated that “diagnostic dataset by searching ICD-10 codes” please indicate which codes of the International Classification of Diseases were used.
  3. ICD acronyms should be explained the first time they appear. Authors may assume that readers know that it refers to the International Classification of Diseases, but this is not always the case.
  4. Bibliographic references should be included when reference is made in the material and methods to the Mayo Clinic score for clinical activity and to the Mayo Clinic 73 endoscopic subscore for endoscopic activity.
  5. To facilitate comparisons, it is suggested that descriptive statistics be presented as mean and SD for quantitative variables or as medians and quartile deviation.
  6. In the material and methods section, the authors state that. “Patients’ characteristics were compared (...)  with the Mann-Whitney 99 test or Student’s t-test.” It should be indicated when each of these tests has been used, i.e., the criteria for using one or the other test.
  7. In the results, the authors present multivariable Hazard Ratios (HR), indicating that a Cox regression has been performed, but nothing is shown in the material and methods. This should be clarified in the article.
  8. With the data in Figure 2, 5-ASA = 5-aminosalicylic acid; IMMs (immunosuppressors) comparisons should be made between the two groups using the log-rank test.
  9. In line, 202 the authors write “conflicting with those published by Salameh ..:” please include the reference. It is indeed at the beginning of the article, but it should be cited again.

Reviewer 2 Report

The authors present the results of retrospective study evaluating long term outcomes of patients with severe UC who underwent treatment with intravenous corticosteroids. They also report predictive factors of relapse in this group of patients. The retrospective nature of the study and the lack of standardised maintenance therapy reduce somewhat the value of the overall observation. Nevertheless, the topic is interesting and the results worthy of attention.

  1. Abstract: line 20-21 please indicate number of patients and % without using fraction.
  2. Poorly described characteristics of the patients at the baseline of the study: no information on whether they had their first or next exacerbation, which drugs were used before the administration of IVCS or CS and whether it was the first or next course of IVCS treatment, etc.
  3. How many patients were followed up for at least 4 years? How many of them were observed for the whole period of study?
  4. The authors state that the assessment of patients was done at 6, 12 and 24 months, but in Table 3 at month 6 they report only 73 patients and not 88 (number of patients included in the study)
  5. Line 125: No explanation of the abbreviation IMM. Please specify which IMMs were used
  6. Were patients observed only until the first exacerbation? Were there patients who had more than one exacerbation during the observation period?
  7. The listing of references should be unified and in line with the editorial requirements

Round 2

Reviewer 1 Report

The authors have corrected all the issues.

Reviewer 2 Report

The authors have responded satisfactorily to the comments which have been raised. I only propose to add that the ICD-10 classification used is in fact the ICD-10 CM version.

I believe that the article can be accepted for publication.